# Cognitive Behaviour Therapy for Depersonalisation Derealisation Disorder (CBT-f-DDD): Study protocol for a randomised controlled feasibility trial

**Georgia McRedmond**[1], **Rafael Gafoor**[1], **Lucy Ring**[2,3], **Nicola Morant**[2], **Joe Perkins**[4], **Nicola Dalrymple**[2,3], **Ana Dumitru**[2,3], **Anthony S. David**[2], **Glyn Lewis**[2], **Elaine C. M. Hunter**[2]*

1 Institute of Clinical Trials and Methodology, Comprehensive Clinical Trials Unit, University College London, London, United Kingdom, 2 Division of Psychiatry, University College London, London, United Kingdom, 3 Camden & Islington NHS Foundation Trust, St Pancras Hospital, London, United Kingdom, 4 Unreal Charity, Bristol, United Kingdom

* e.hunter@ucl.ac.uk

**Data Availability Statement:** We are not able to publicly share the data of this study as these

## Abstract

### Background

Depersonalisation-Derealisation Disorder (DDD) is a distressing mental health condition which causes individuals to have a sense of 'unreality' or detachment about themselves and/or the world around them. DDD is chronically under-researched, and as a result, under-diagnosed, with a population prevalence of about 1%. In systematic reviews, Cognitive Behavioural Therapy (CBT) has been found to be the only intervention with significant clinical impact on alleviating the symptoms of DDD. However, previous studies have suffered from small sample sizes, reliance on expert clinicians to provide therapy and narrow population demographics. This feasibility randomised controlled trial aims to provide more robust evidence for the treatment efficacy of CBT in DDD.

### Methods

The study aims to recruit 40 participants from two NHS trusts, 20 per arm from two community Mental Health NHS services in London. The intervention group will receive 12–24 individual CBT sessions over a 6-month period from CBT therapists following specialist training for DDD. The control group will receive Treatment as Usual. We will assess the feasibility of a future RCT through measuring the acceptability of the intervention, and assessing our ability to recruit, retain and randomise participants. We will calculate the correlation of scores on the Cambridge Depersonalisation Scale, its baseline standard deviation, assess the magnitude/direction of change and characterise the uncertainty in the outcome scores and the probability that the results have been obtained by chance.

contain personally identifiable information and we are restricted from sharing these by our Research Ethics Committee. Requests for data access can be sent to Department of Psychiatry 6/F, Maple House, 149, Tottenham Court Rd, London, UK, W1T 7NF via DoP.admin@ucl.ac.uk.

**Funding:** This project is funded by the National Institute for Health and Care Research (NIHR) under its Research for Patient Benefit (RfPB) Programme (Grant Reference Number NIHR202201). The views expressed are those of the authors and not necessarily those of the NIHR or the Department of Health and Social Care. The study is sponsored by Camden & Islington NHS Foundation trust, sponsor's number: H-1809. Additional support is being provided by University College London and Barnet, Enfield, and Haringey NHS Foundation Trust. The sponsor is responsible for the study design, conduct of the study, data analysis and interpretation, writing of the manuscripts and dissemination of results. The sponsor will make the final decisions regarding any of these aspects of the study. Sponsor's contact details: sponsor.noclor@nhs.net Camden & Islington NHS Foundation Trust of St. Pancras Hospital, 4 St. Pancras Way, London, United Kingdom, NW1 0PE. 0203 317 3034.

**Competing interests:** The authors have declared that no competing interests exist

**Abbreviations:** BEH, Barnet, Enfield and Haringey; CBT, cognitive behavioural therapy; CBT-f-DDD, Cognitive Behavioural Therapy for Depersonalisation Derealisation Disorder; CCG, Clinical Commissioning Group; CDS, Cambridge Depersonalisation Scale; CI, Chief Investigator; CIS-R, Clinical Interview Schedule-Revised; CTS-R, Cognitive Therapy Scale-Revised; C&I, Camden and Islington; DDD, Depersonalisation Derealisation disorder; DES, Dissociative Experiences Scale; GAD-7, Generalised Anxiety Disorder Assessment 7; IDMEC, Independent Data Monitoring and Ethics Committee; NHS, National Health Service; PAG, Patient Advisory Group; PHQ-9, Patient health questionnaire 9; RA, Research Assistant; SSRI, Selective Serotonin Reuptake Inhibitor; TAU, Treatment as Usual; TMG, Trial Management Group; TSC, Trial Steering Committee; UCL, University College London; WSAS, Work and Social Adjustment Scale.

## Discussion

The outputs of this trial will guide whether a definite RCT is feasible and acceptable, for both the clinician and participant.

## Trial registration

The ISRCTN registration number is ISRCTN97686121(https://doi.org/10.1186/ISRCTN97686121).

## Background

Depersonalisation Derealisation Disorder (DDD) is a distressing mental health condition where a person has a profound sense of disconnection and unreality about themselves and/or the world [1]. Systematic reviews estimate the prevalence of DDD in community surveys at around 1%, which is similar to rates of schizophrenia and obsessive-compulsive disorder [2–4]. The prevalence of DDD amongst mental health service users is higher at approximately 20%, usually in combination with other disorders [3,4]. However, there is a gap between true prevalence rate and clinical diagnosis; chronic under-diagnosis of DDD, often coming after delays of several years, has contributed to the widely held, but erroneous, assumption that DDD is rare [5]. Consequently, there has been underfunding of research into effective treatments. There is currently no National Institute of Care Excellence (NICE) guidance on treatments for DDD [6]. Recent lobbying of parliament has achieved formal acknowledgement of the urgent need for more research [7].

Cognitive behavioural therapy (CBT) is the most widespread psychological therapy used in the NHS with effectiveness demonstrated across a wide range of conditions [6]. CBT seeks to identify cognitive (i.e., thoughts and thinking processes), emotional and behavioural factors that might perpetuate symptoms and seeks to find more helpful alternatives [8]. Although the basic cognitive and behavioural theory of emotion and behaviour remains the same, adaptations of CBT to different disorders consider the most relevant areas to address in therapy [8,9].

In systematic reviews of the existing evidence of treatment for DDD, the strongest evidence is for CBT that has been adapted for the specific condition (CBT-f-DDD), with little benefit demonstrated from a variety of medications and other therapy modalities [10]. Wang et al.,'s [9] recent systematic review comprehensively covered both psychiatric and therapeutic modalities for treating DDD concluding that the evidence was poor, heterogenous and cannot support clinical treatment. The main treatment modalities discussed included medication primarily the use of lamotrigine and selective serotonin reuptake inhibitors (SSRIs) [11–13]; neuromodulation of the temporo-parietal junction and ventrolateral prefrontal cortex through repetitive transcranial magnetic stimulation [14]; and CBT methods as described in this study [15–17].

There have been two uncontrolled studies of consecutive referrals of patients with DDD to a tertiary specialist clinic [15,16]. In the first of these, 21 participants, with longstanding severe symptoms which had not responded to medication or previous psychological therapies, were treated with a median of 11 individual adapted CBT-f- DDD sessions (IQR: 7 to 18). There was a statistically significant reduction in state DDD symptom severity (Effect Size = 0.37, 95%; [C.I: 0.12 to 0.53]) with 29% of participants no longer meeting diagnostic criteria at the end of therapy. Additionally, participants experienced a reduction in the severity of general dissociation (Effect Size = 0.34; [95% C.I: 0.10 to 0.51]). A recent analysis of outcomes of 36

participants from the same clinic has replicated these initial findings [15]. A mean decrease in trait DDD symptoms (Mean = -33.67; [95% C.I: -45.14 to -22.20]) was observed after a median of 16 sessions of CBT (IQR: 10 to 16), with additional statistically significant reductions in anxiety and depression symptoms during the treatment period relative to the waitlist period.

There are several methodological limitations to these studies: participants were not randomised, participants were only recruited from one tertiary specialist clinic and therefore may not be typical of those in generic primary and secondary care settings, and a small number of clinicians delivered the intervention. It is not possible to make concrete inference about the effectiveness of CBT for DDD or about its generalisability using these limited data.

## Summary and research questions

Given the relative paucity of robust information about the effectiveness of CBT for DDD currently available, and not having the necessary parameters with which to conduct a sample size calculation for a fully powered definitive RCT, we are carrying out a feasibility study to assess the feasibility and acceptability of carrying out a future RCT of CBT for DDD.

## Methods/Design

This project will use a two-arm parallel group (single blinded to assessor) individually randomised design consisting of active CBT treatment versus Treatment As Usual (TAU). Participants will undertake an eligibility assessment (T0) and will be assessed again prior to commencement of treatment (T1). Participants randomised to the treatment condition will receive a minimum of 12 weekly CBT sessions (up to a maximum of 24) during a 6-month intervention window. Once the intervention is complete, all participants will be assessed again (T2, 6 months after the eligibility assessment). The final follow-up assessment will occur for all participants nine months after the eligibility assessment (T3).

## Aims and objectives

The aims of the study are to:

1. Investigate the ability to recruit, randomise and retain participants through the study period,

2. Define the parameters to inform the sample size calculation for a future efficacy trial (RCT) (i.e., to obtain the correlation coefficient of scores from the Cambridge Depersonalisation Scale (CDS) at baseline and at follow-up) as well as the pooled standard deviation of the CDS at baseline,

3. To carry out preliminary inferential analyses of the efficacy of the therapy to check if a signal has been detected in the treatment arm versus the control arm (so as to inform a subsequent larger RCT if appropriate). These analyses may include estimating the magnitude and direction of the treatment effect as well as obtaining an estimate of the uncertainty in the estimate.

4. To perform a within-trial economic evaluation to compare the costs and outcomes of the intervention group versus the standard care group,

5. Disseminate specialist CBT-f-DDD skills to generic NHS clinicians,

6. Determine whether CBT-f-DDD can be conducted within typical NHS settings with generic CBT therapists given a brief training in DDD.

**Participants.** The study will aim to recruit 40 participants, 20 participants in each arm. All participants will be aged between 16–75, living in London, and have symptoms of Depersonalisation and/or Derealisation currently meeting DSM-V (1) criteria for DDD. This will be assessed by the research assistant (RA) in their initial (T0) screening interview by asking the participant the following two questions, which make up part of the diagnostic criteria in the DSM-V (1):

"Over the last two weeks, how often have you had:

a. Experiences of unreality, detachment, or being an outside observer with respect to one's thoughts, feelings, sensations, body, or actions (e.g., perceptual alterations, distorted sense of time, unreal or absent self, emotional and/or physical numbing). [Depersonalisation] 0 = not at all; 1 = several days; 2 = more than half the days; 3 = nearly every day"

b. Experiences of unreality or detachment with respect to surroundings (e.g., individuals or objects are experienced as unreal, dreamlike, foggy, lifeless, or visually distorted). [Derealisation] 0 = not at all; 1 = several days; 2 = more than half the days; 3 = nearly every day."

The total score will be calculated as follows:

1. Total score = (a + b).

   a. Scores <3 = mild-moderate DDD.

   b. Scores > = 3 = moderate-severe DDD.

Other DSM-V criteria will be assessed including intact reality testing, that the symptoms result in significant distress and/or functional impairment, and that the DDD symptoms cannot be better explained by another condition. Participants will be excluded if they have any of the following co-morbid diagnoses: any psychosis spectrum disorder, substance dependence, intellectual disability, Post Traumatic Stress Disorder (PTSD), or cognitive impairment due to head injury or organic disorder. Comorbidity with anxiety and/or depressive disorders will not be an exclusion criterion.

Additionally, participants will be excluded from the study if they have previously received CBT specifically for the treatment of DDD; have a score below clinical severity (i.e. total score < 3) on the diagnostic questions; cannot provide informed consent; do not have sufficient proficiency in English to engage in CBT; require special communication needs arrangements; are not registered with a GP practice or are involved in any other mental or physical research studies (including drug trials).

*Diversity and Inclusion.* To increase the likelihood that the study sources a representative sample, recruitment will take place across two London mental health trusts covering six boroughs. Camden and Islington NHS Foundation Trust (C&I) is an inner London trust which had over 45,600 service users in 2020 [18]. The trust covers the boroughs of Camden, Islington, and Kingston. Islington is the most densely populated local authority area in England and Wales and is among the poorest 20% of neighbourhoods in England [19]. Barnet, Enfield and Haringey NHS Foundation Trust (BEH) is an outer London trust which had 146,000 service users in 2022 [20]. Approximately a third of individuals across all six boroughs are residents from black, Asian or minority ethnic groups [19,21–25]. We will be recruiting from primary, secondary and tertiary care services across these Trusts.

**Sample size.** There is no formal sample size calculation, or priori power calculation. Instead, a minimum of 24 participants, 12 in each arm, has been suggested for pilot studies

[26]. This minimum recruitment strategy requires recruiting 29 eligible participants in total (encompassing a nominal 20% dropout rate). This study aims to recruit 20 participants in each arm who complete. We believe this number will give us a final sample large (and sufficiently diverse) enough to assess the practicalities of recruitment, retention, randomisation, and intervention delivery, as well as to obtain the necessary data for the statistical inferences required.

A previous naturalistic self-controlled cross-over study found the effect size between the treatment period and the waiting period to be 0.52 (improvement in favour of treatment). This translated into a decrease of 36 points on the CDS Scale. Using this change score, a correlation of 0.9 which was observed for intra-participant correlation, and the standard deviation of the CDS at each phase of the crossover study, we anticipated that the required sample size would be 29 participants per arm (using the ANCOVA method). Our planned recruitment of 20 per arm provides sufficient power to provide information on the magnitude, direction, and precision of the treatment effect from the feasibility study to power (with sufficient precision) a subsequent definitive RCT if appropriate [15].

To meet our relatively conservative aim of 20 participants completing the trial in each arm we will need a recruitment rate of 5–10 participants a month during the 6-month recruitment period. Historical data demonstrates a referral rate of one referral per week, which would be adequate for the proposed sample size.

**Intervention.** CBT for DDD aims to reduce distress associated with DDD symptoms by changing catastrophic attributions through general and personalised psychoeducation; developing a shared formulation of difficulties by linking DDD symptoms to anxiety, mood, and/or past traumas; enhancing coping strategies; and by using cognitive restructuring techniques to modify unhelpful appraisals. CBT-f-DDD theorises that DDD symptoms are reduced by altering the maintenance cycle and associated distress.

The intervention will be delivered in addition to the participant's TAU over a six-month period. The CBT-f-DDD protocol and training manual developed by ECMH will be provided to therapists during training, and each therapist will adjust these as appropriately determined to the individual needs of the client. The intervention comprises the following components (see Table 1).

**Treatment as usual condition.** For some participants, the TAU will involve contact with a therapist through their mental health team or medication prescribed by their GP and/or mental health team. The TAU will vary from individual to individual but may consist of waitlist, short-term psychological therapies, or prescription of antidepressants such as SSRIs. Participants will be asked to keep a record of the type and frequency of their appointments with clinical services and treatments offered. These data will be supplemented with information provided by their key worker or through clinical note screening.

## Procedure

**Phase 1: Recruitment and Training.** Participants and therapists will be recruited via two NHS Mental Health Trusts:

- An inner London mental health trust (Camden and Islington NHS Foundation Trust (C&I))

- An outer London mental health trust (Barnet, Enfield, and Haringey Mental Health NHS Trust (BEH))

The two NHS Trusts will be our primary recruitment sites, and clinicians will be aiding in identification and recruitment as well as becoming the first point of contact for potential participants. Additionally, the study will be advertised on both Trust's websites as well as on the

**Table 1. CBT-f-DDD intervention components.**

| Engagement and psychoeducation about DDD |
| --- |
| • Managing expectations and goal setting |
| • General and individualised psychoeducation about DDD |
| **Developing the shared formulation** |
| • Diary keeping and analysis |
| • Identifying external and/or internal triggers |
| • CBT vicious cycle of recent, typical incident |
| **Cognitive strategies: content and process** |
| • Restructuring beliefs about DDD |
| • Recognising thinking biases |
| • Working with cognitive processes to reduce rumination, worry and hypervigilance |
| **Emotional regulation strategies** |
| • Anxiety and mood management strategies |
| • Grounding strategies |
| **Behavioural Interventions** |
| • Psychoeducation about safety seeking behaviours |
| • Behavioural activation |
| • Behavioural experiments |
| • Graded exposure |
| **Working with common co-morbid conditions triggering DDD** |
| • Anxiety disorders, low mood, low self-esteem |
| • Working with negative core beliefs/schemas, procrastination, perfectionism |
| • Substance use |
| **Working with Issues Related to Onset** |
| • Trauma-focused CBT imaginal exposure |
| • Dealing with physical reminders of onset |
| • Chair work where the client uses different chairs to represent different aspects of themselves to foster communication and insights from taking various perspectives |
| **Working with Predisposing Factors** |
| • Working with past trauma or life adversity |
| **Staying Well Plans** |

Unreal Charity's website (https://www.unrealuk.org/) [27], which provides a route to self-refer to the study.

If self-referred, the potential participant will be provided with a Participant Information Sheet (PIS) and follow the route to obtaining informed consent (unless they satisfy the exclusion criteria or do not fulfil the inclusion criteria). No information about participants will be collected and no research activities will be held within Unreal. Unreal [27] will be used solely as a participant identification centre whereby the participants recruited will be advised that they will be referred to one of the NHS trusts for the duration of their participation in the study. We will involve the externally recruited participants' GPs/mental health teams to gain information relevant to risk, current prescribed medication, and use of services.

Informed consent will be collected at T0 by a RA, prior to baseline interview and after the eligibility screening. Participants will be randomised following the baseline interview (T0) and collection of baseline data (see Fig 1).

*Training of Therapists*. Training of therapists will take place in half-day workshops. All therapists involved will be qualified and have experience with CBT. We will collect data about whether therapists included in the study are accredited with the British Association of

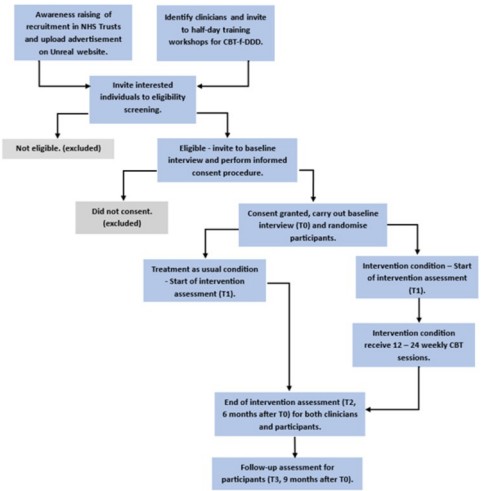

**Fig 1. Trial flowchart.**

Behavioural and Cognitive Psychotherapists (BABCP) and their past qualifications and years of experience with CBT. The training workshops will aim to help therapists understand DDD and demonstrate the adaptation of existing CBT skills to work with DDD using clinical case examples. All therapists will receive a hard copy of the training manual with additional materials and reading. As this is a feasibility study, we will also be examining whether a half-day training would be sufficient to supply these skills and knowledge.

There is an agreement with the relevant C&I and BEH Heads of Services that therapists across the trusts will attend the training in order to be able to take on one or more cases of DDD clients. We envisage having a relatively large number of therapists with a small trial caseload to reduce the burden on services and to maximise the number of therapists who will benefit from training and specialist supervision.

**Phase 2: Treatment Phase.** Prior to commencing therapy and one month after randomisation, the participants in both arms will be assessed again (T1). Those in the intervention arm will receive a minimum of 12 (maximum of 24) 1-hour-long CBT-f-DDD sessions. The location of these sessions will depend on participant and/or therapist preference and can be conducted face-to-face or via videoconferencing. Meta-analyses comparing the efficacy of psychological therapies conducted face-to-face versus through videoconferencing has shown similar symptomatic improvement and client satisfaction scores across conditions [28,29]. Additionally, a recent RCT examining online versus in-person CBT for anxiety conditions did not find a significant difference in outcome measures between treatment conditions [30]. It is not expected that this study will find any differences between the two conditions but will collect the modality used for monitoring purposes. The CI [ECMH] will provide fortnightly group specialist supervision to therapists delivering the intervention. For those in the TAU arm, there will be a record kept of the type and frequency of their appointments with clinical services and the treatments offered.

At the end of the intervention period, all participants will complete an end of treatment assessment (T2). Therapists will also be invited to participate in a semi-structured qualitative interview to explore their views on the acceptability, feasibility and perceived impact of training, therapy delivery and supervision once they have completed work with all their clients (T2).

**Phase 3: Follow-up.** The follow-up period will begin nine months after randomisation and three months after the intervention ended. Participants from both arms of the trial will be

invited to the follow-up assessment (T3). For those in the intervention arm, the follow-up will additionally involve a semi-structured qualitative interview to record their experiences of treatment and its impact. We will attempt to contact those who were lost to follow-up to try to understand their reasons for dropping out. Particularly, we are interested in whether the loss of contact is related to intervention acceptability issues. Upon completion of their participation in the study, participants will receive a £10 Love2Shop voucher.

Dissemination of the findings will take place within Unreal, NHS partners, University College London (UCL), and the wider academic/CBT community. This will be achieved through conference presentations, and written peer-reviewed publications. Participants will have the option of requesting study results upon completion by ticking a box in the consent form. Unreal as a co-applicant on this study will assist the dissemination of materials to the lay public across various platforms. There will be a plain English summary published on the charity website and the results of the study will be included in their charity newsletter which is circulated to 1100 recipients [31]. Additionally, a plain English thread will be posted on their X account to reach both wider academic audiences and clinical populations.

**Data collection.** This study will focus on assessing the feasibility of future studies and the acceptability of CBT-f-DDD as an intervention (see S1 Appendix). To assess feasibility, resources needed to complete the feasibility trial will be recorded including:

- Length of time required for RA to complete assessments and to collect and analyse data, collected from RA time sheets.

- CBT attendance rate, collected from electronic patient records.

- The amount and nature of supervision required, collected from supervision notes.

  Recruitment data collected throughout the study will also aid in assessing the feasibility:

- Number of referrals from each source.

- Number of ineligible referrals.

- Reasons for ineligibility.

- Rates and reasons for refusal to be included in the study.

- Rates of attrition.

- Reasons for withdrawing from the study.

We will measure the participants' acceptability of the intervention and study procedure, and the therapists' acceptability of the intervention. Attrition rates, and the number of therapy dropouts will also be collected in the measurement of acceptability.

Participants' acceptability of the intervention as well as the assessment procedures and measures will be collected through an end of intervention questionnaire at T2 and semi-structured qualitative interviews at T3 carried out by RAs. Participants' acceptability of CBT-f-DDD will be measured through questions regarding which aspects of the treatment they found helpful and unhelpful, their perceived impact of the intervention, and their satisfaction with the intervention and therapists. Additionally, a global measure of change question with a binary response will be asked:

"Do you feel better? Yes/No"

Therapists' acceptability of the intervention will be measured through semi-structured qualitative interviews at T2. This will measure therapist satisfaction with training and supervision, their confidence in delivering the therapy, their views of the intervention's value, and the

perceived impact and feasibility of delivery in standard clinical settings. The therapists' adherence to the intervention protocol will also be examined in measuring the acceptability of the intervention through:

- Participant's engagement with CBT-f-DDD and homework tasks using a Likert scale for clinicians to complete,

- A random sample analysis of 10% of audio-recorded therapy sessions evaluated through standardised CBT protocols for adherence to the CBT model using the Cognitive Therapy Scale-Revised (CTS-R) and an adapted CBT-f-DDD checklist (see S2 Appendix).

Finally, we will make a health economics evaluation of the costs and outcomes of the intervention group versus the standard care group. This will include:

- Baseline data on health service usage using The Client Service Receipt Inventory [32], which collects information of participants' use of health and social care services, living situation, income, employment and benefits, NHS and private costs.

- Quality of life will be assessed using the EQ5D5L [33] at baseline (T0 assessment) after treatment (T2 assessment) and at 9-month follow up (T3 assessment).

*Clinical Data*. Demographic and clinical data will be collected at baseline interview (T0). The demographic data will include age, gender, ethnicity, education, and employment status. Additionally, clinical demographics will be collected including age of onset of DDD, duration, co-morbid diagnoses, previous treatments for DDD, and type and dosage of current medication. The demographic and clinical data collected on the sample will be used to check the representativeness of the sample.

Secondary clinical outcomes will also be measured to inform future trial design as advised by the CONSORT guidance on feasibility RCTs [34] (see Table 2). These measures will provide preliminary data on the correlation between the baseline and final outcome data, the magnitude and direction of the difference between the two groups and the pooled standard deviation of the primary outcome at baseline. They will also provide data on the feasibility of randomisation, recruitment, retention, and acceptability of the intervention. These data will allow for an evaluation of carrying out a future RCT.

*Data Management*. All study data will be kept in electronically and physically secure environments and will be pseudo anonymised via the allocation of a unique alphanumeric identifier. The participants will have individual meetings with a research assistant, allowing participants the privacy and opportunity to discuss any concerns and participants will be able to leave the study at any time, without giving a reason. Additionally, participants will be invited to give feedback on their experiences at all stages of the study.

An independent Participant Advisory Group (PAG), consisting of individuals with lived experience of the condition, will provide feedback on every stage of the study. The PAG will meet at key stages throughout the study to be involved with recruitment, intervention, analysis, and interpretation. Activities of the PAG will include reviewing study materials; input into designing qualitative interview topic guides; and providing a lived experience perspective on recruitment and trial processes.

The Independent Data Monitoring and Ethics Committee (IDMEC) and the Trial Steering Committee (TSC) will comprise of independent investigators operating on behalf of the funders who will meet as they see fit. The IDMEC will have untrammelled access to the unblinded data (if so requested) and will report the results to the TSC. Additionally, the Trial Management Group (TMG) will meet monthly to discuss trial management and proceedings. Trial

**Table 2. Clinical outcomes and measures.**

| Outcome | Measure | Description |
|---------|---------|-------------|
| Depersonalisation | The Cambridge Depersonalisation Scale (CDS) [35] | The CDS is a comprehensive 29 item assessment of depersonalisation over the last 6 months, with each item rated by two Likert scales for frequency and duration of experience. The CDS addresses symptoms consistently reported for individuals experiencing depersonalisation including altered experiences in several sensory modalities and in cognition [35]. Scores range from 0–290. |
| Dissociation | Dissociative Experiences Scale (DES) [36] | The DES is a self-report, 28-item questionnaire which asks the responder about the percentage of time a range of dissociative experiences happen in their adult life. The questionnaire includes three subscales: 'Depersonalisation/Derealisation'; Dissociative Amnesia; and Absorption [36]. |
| Depression | Patient Health Questionnaire 9 (PHQ-9) [37] | The PHQ-9 is the depression module of a larger diagnostic instrument for common mental disorders. The participant is asked to rank symptom frequency of each of the nine DSM-IV criteria for depression on a four-point scale from "0 (Not at all)" to "3 (Nearly every day)". Scores range from 0–27. |
| Anxiety | The Generalised Anxiety Disorder Assessment 7 (GAD-7) [38] | The GAD-7 is a seven-item scale reflecting the criteria for generalised anxiety disorder in the DSM-IV. The participant is asked to rank symptom frequency on a four-point scale from "0 (Not at all)" to "3 (Nearly every day)". Scores range from 0–21. |
| Co-Morbidity | The Clinical Interview Schedule–Revised (CIS-R) [39] | The CIS-R evaluates common mental disorders through assessing the frequency of symptoms in the last month across 14 symptom groups. The symptoms are all ranked from 0–4, with higher ratings indicating greater symptomatology [40]. Scores range from 0–56. |
| Functioning | The Work and Social Adjustment Scale (WSAS) [41] | The WSAS is a self-report measure of impaired functioning with five items. The WSAS measures the client's perceived work and social functional impairments from "0 (Not at all impaired)" to "8 (Severely impaired)". Scores range from 0–40. |

steering groups will ensure the trial adheres to the study protocol, ensures patient safety, and progresses as communicated to the sponsors and funders.

**Ethics approval and consent to participate.** Full Health Research Authority (HRA) and NHS Research Ethics Committee (REC) approval has been granted (IRAS ID: 314923; 22/LO/0318).

**Randomisation.** Randomisation lists created by an independent statistician prior to recruitment to the study, and they will be delivered to the recruitment sites by sealed envelope method. All participants and study staff will not be blinded apart from the principal statistician [RG] who will be unblinded at the end of the study after data lock has occurred.

Unblinded results will be discussed in closed sessions from which the trial statistician will be excluded. Unblinded interim results will be generated (if necessary) by the independent statistician and made available to the IDMEC for each sitting if requested by the Chair of the committee.

## Analyses

We will seek evidence of the feasibility of conducting a larger subsequent RCT by examining the following parameters:

- Power calculation: We will determine the pooled standard deviation for the baseline score of the primary outcome variable, as well as the correlation of the primary outcome at baseline and at follow-up (T3). These data, along with the mean clinically important difference (MCID), will be used in the sample size calculation for a fully powered subsequent study.

- Recruitment: Number of referrals from each source, numbers ineligible, reasons for ineligibility, rates, and reasons for refusal to be included in the study.

- Randomisation: whether groups are balanced and randomisation possible in this environment.

- Withdrawals: Attrition rates and reasons for withdrawing throughout the study, follow-up rates.

Additionally, to fulfil the secondary objectives, descriptive analyses will be performed on the following outcomes: resources needed to complete CBT-f-DDD; health economics evaluation; therapist adherence to CBT-f-DDD; sample characteristics; and acceptability outcomes.

Qualitative data from interviews with therapists and participants will be transcribed and analysed using thematic analysis within NVivo software (supervised by co-applicant Dr Morant). This will be guided by Braun & Clarke's six stage process [42], and conducted collaboratively to include the perspectives of clinicians and those with lived experience within the research team. Interviews will be recorded using an encrypted device for the audio recording, and data will be stored securely within the NHS. Transcription of this audio recording will be done by a professional external agency which has an over-arching contract with UCL that covers all GDPR issues (as outlined in the collaboration agreement between UCL and C&I). These audio files will be destroyed at the end of the project.

As well as providing an in-depth exploration of treatment experiences and perceived impacts (from both provider and recipient perspectives), analysis will be structured to produce clear summaries of feasibility and acceptability issues as reported by therapists and participants. These analyses will inform therapy and trial process modifications in advance of any subsequent definitive trial.

The final dataset will only be accessible to the minimum number of individuals necessary for quality control, audit, and analysis, which will be members of the research team and potentially members of the TSC and IDMEC. Given these data could have the potential for reverse deanonymisation of participants, the final study dataset will not be available to others beyond the immediate research team and committees. However, the details of the statistical analysis and the analysis of transcripts will be made more widely available.

## Adverse events

This is not a high-risk study and we do not anticipate any serious adverse events occurring within the study. However, we have undertaken a Research Risk Assessment and a Data Protection Impact Assessment to identify potential risks to mitigate against these as much as possible in advance. From this a Risk Management Plan was put in place to inform RAs of the appropriate procedure when/if there is a disclosure of suicidal ideation made. Finally, we are aware that within any therapy there is the potential discussion of past difficult events which may cause distress to the participant. All interventions will be carried out by qualified and trained NHS clinicians who will also receive regular, specialised supervision by ECMH.

## Discussion

This paper presents the protocol for a feasibility study investigating a cognitive behavioural therapy intervention for individuals with DDD. This research came about after two clinical audits from a national specialist NHS service indicated that CBT adapted for DDD might be of benefit. However, these were small scale, from a specialist service and participants had not been randomised, so the next step in both research and treatment provision in this clinical population was to identify whether a future RCT is feasible and whether the intervention,

CBT-f-DDD, is acceptable to both clinicians and clients. The findings of the feasibility study will aid in establishing the key parameters for a future RCT trial (if appropriate). Additionally, this feasibility paper will also help in dissemination of information about CBT-f-DDD for up to 20 clinicians, hopefully increasing awareness and improving accuracy in diagnoses of DDD for both NHS trusts involved.

Protocol version: V6.6 September 2023.

## Supporting information

**S1 Checklist. SPIRIT-Outcomes 2022 checklist (for combined completion of SPIRIT 2013 and SPIRITOutcomes 2022 items)[a].**
(PDF)

**S1 Appendix. Table A1.** Spirit Figure.
(DOCX)

**S2 Appendix. Figure B1.** Adapted CBT-f-DDD checklist.
(TIF)

**S1 File.**
(DOCX)

## Acknowledgments

**Declarations**

We acknowledge the funder of the study (NIHR) and the generous participation of the patients who have agreed to take part in this study.

## Author Contributions

**Conceptualization:** Rafael Gafoor, Nicola Morant, Joe Perkins, Anthony S. David, Glyn Lewis, Elaine C. M. Hunter.

**Data curation:** Georgia McRedmond, Rafael Gafoor, Lucy Ring, Nicola Morant, Nicola Dalrymple, Ana Dumitru.

**Formal analysis:** Rafael Gafoor, Lucy Ring, Nicola Morant, Nicola Dalrymple.

**Funding acquisition:** Rafael Gafoor, Nicola Morant, Joe Perkins, Anthony S. David, Glyn Lewis, Elaine C. M. Hunter.

**Investigation:** Rafael Gafoor, Lucy Ring, Nicola Morant, Nicola Dalrymple, Ana Dumitru, Anthony S. David, Glyn Lewis, Elaine C. M. Hunter.

**Methodology:** Rafael Gafoor, Lucy Ring, Nicola Morant, Joe Perkins, Ana Dumitru, Anthony S. David, Glyn Lewis, Elaine C. M. Hunter.

**Project administration:** Lucy Ring, Nicola Dalrymple, Ana Dumitru, Elaine C. M. Hunter.

**Resources:** Ana Dumitru.

**Software:** Rafael Gafoor, Elaine C. M. Hunter.

**Supervision:** Rafael Gafoor, Nicola Morant, Anthony S. David, Glyn Lewis.

**Writing – original draft:** Georgia McRedmond, Rafael Gafoor, Lucy Ring, Elaine C. M. Hunter.

**Writing – review & editing:** Rafael Gafoor, Nicola Morant, Joe Perkins, Nicola Dalrymple, Ana Dumitru, Anthony S. David, Glyn Lewis, Elaine C. M. Hunter.

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
