## [Decision Letter · Decision Letter 0]

25 Mar 2024

PONE-D-24-01397Cognitive Behaviour Therapy for Depersonalisation Derealisation Disorder(CBT-f-DDD): Study Protocol for a Randomised Controlled Feasibility Trial

PLOS ONE

Dear Dr. Hunter,

Thank you for submitting your manuscript to PLOS ONE. After careful consideration, we feel that it has merit but does not fully meet PLOS ONE’s publication criteria as it currently stands. Therefore, we invite you to submit a revised version of the manuscript that addresses the points raised during the review process.

Adjust citation method to adhere to journal guidelines.

Clarify the term "talking therapy" and specify included treatments like counseling or CBT.

Include the number of sessions in the initial CBT study on DDD.

Expand literature review to cover various DDD treatments comprehensively.

Elaborate on the process of examining diagnostic criteria, including details on diagnostic interviews and personnel involved.

Provide more information about "treatment as usual."

Specify the qualitative method used for data analysis.

Discuss measures taken to compare in-person and online therapy methods, considering potential differences and limitations.

Explain the duration of treatment sessions.

We look forward to receiving your revised manuscript.

Kind regards,

Muhammad Shahzad Aslam, Ph.D.,M.Phil., Pharm-D

Academic Editor

PLOS ONE

Additional Editor Comments:

Adjust citation method to adhere to journal guidelines.

Clarify the term "talking therapy" and specify included treatments like counseling or CBT.

Include the number of sessions in the initial CBT study on DDD.

Expand literature review to cover various DDD treatments comprehensively.

Elaborate on the process of examining diagnostic criteria, including details on diagnostic interviews and personnel involved.

Provide more information about "treatment as usual."

Specify the qualitative method used for data analysis.

Discuss measures taken to compare in-person and online therapy methods, considering potential differences and limitations.

Explain the duration of treatment sessions.

Reviewers' comments:

Reviewer's Responses to Questions

**Comments to the Author**

1. Does the manuscript provide a valid rationale for the proposed study, with clearly identified and justified research questions?

Reviewer #1: Yes

Reviewer #2: Yes

2. Is the protocol technically sound and planned in a manner that will lead to a meaningful outcome and allow testing the stated hypotheses?

Reviewer #1: Yes

Reviewer #2: Yes

3. Is the methodology feasible and described in sufficient detail to allow the work to be replicable?

Reviewer #1: Yes

Reviewer #2: Yes

4. Have the authors described where all data underlying the findings will be made available when the study is complete?

Reviewer #1: No

Reviewer #2: Yes

5. Is the manuscript presented in an intelligible fashion and written in standard English?

Reviewer #1: Yes

Reviewer #2: Yes

6. Review Comments to the Author

You may also provide optional suggestions and comments to authors that they might find helpful in planning their study.

Reviewer #1: Greetings,

I am grateful for the opportunity to review this valuable manuscript. I have read through the paper you submitted in detail. I have provided feedback to enhance the quality and impact of your article, and in order to better understanding of readers. I'm eager to review the revised version after the changes are implemented.

Introduction:

1. The method of citing sources should follow the journal's guidelines and needs to be modified.

2. Since the term talking therapy is not common in many countries, Include information on what kind of treatments it includes. Counseling? Self-help? CBT?

3. Please provide information about the number of sessions in the first research you mentioned about CBT in DDD.

4. It could benefit from a more comprehensive review of relevant literature on different treatments of DDD.

Method and result:

1. consider providing more detail on the process of examining diagnostic criteria. Is there a diagnostic interview? by whom

2. Please provide more details about treatment as usual.

3. Considering that qualitative analysis is used in the present study, please indicate which qualitative method you will use to analyze the data.

4. Considering that both in-person and online methods are used, what measures have you taken to compare these two methods? These two types of psychotherapy can achieve different results and have their own limitations.

5. Please explain about the duration of treatment sessions.

Regards,

Reviewer #2: "Cognitive Behaviour Therapy for Depersonalisation Derealisation Disorder (CBT-f-DDD): Study Protocol for a Randomised Controlled Feasibility Trial," it appears to be a thorough and well-structured study protocol aimed at addressing a significant gap in mental health research. The study is designed to evaluate the feasibility of a full-scale Randomised Controlled Trial (RCT) for Cognitive Behaviour Therapy tailored for Depersonalisation Derealisation Disorder (DDD), a condition that remains under-researched and under-diagnosed.

Here are are some recommendations for the editor regarding this manuscript:

Strengths:

1. Clear and Significant Rationale: The background section effectively highlights the importance of researching DDD and the potential of CBT as a treatment, establishing a strong rationale for the study.

2. Comprehensive Methodology: The study's methodology is well-designed, detailing participant recruitment, intervention procedures, data collection, and planned analyses. This thoroughness enhances the protocol's feasibility and potential impact.

3. Ethical Considerations and Transparency: The protocol includes detailed information on ethical approvals, consent processes, and data management, reflecting a commitment to ethical standards and participant safety.

4. Multidimensional Data Analysis: The plan to use both quantitative and qualitative data analysis enriches the study's approach to understanding the feasibility and acceptability of CBT-f-DDD, providing a holistic view of the intervention's potential.

Recommendations for Improvement:

1. Sample Size Justification: While the protocol outlines the rationale for the chosen sample size based on feasibility, further justification regarding its adequacy for preliminary efficacy analysis might strengthen the proposal. Exploring the expected effect size based on prior studies could offer additional insight into the chosen sample size.

2. Diversity and Inclusion: Given the mention of narrow population demographics in previous studies, it would be beneficial to outline specific strategies for ensuring a diverse participant pool, addressing potential barriers to participation for underrepresented groups.

3. Clarification on Control Group Treatment:*The protocol mentions that the control group will receive Treatment as Usual (TAU). Clarifying what TAU entails within the two NHS trusts and how it differs from the intervention could provide valuable context for interpreting future results.

4. Long-Term Follow-Up Plans: Although the study focuses on short-term feasibility and acceptability, discussing plans or potential for long-term follow-up could enhance understanding of the sustained impact of CBT-f-DDD, addressing whether symptom improvements are maintained over time.

5. Dissemination Strategy: Expanding on the dissemination plans to include patient and public involvement strategies might increase the study's impact. Outlining how findings will be communicated to participants, patient groups, and the wider public could foster engagement and awareness.

Conclusion:

The study protocol presents a well-considered approach to a significant gap in mental health treatment research. With minor enhancements to strengthen the rationale for sample size, ensure diversity, clarify control group treatment, consider long-term impacts, and expand dissemination plans, this protocol has the potential to significantly contribute to the field and pave the way for a larger-scale RCT.

7. PLOS authors have the option to publish the peer review history of their article (what does this mean?). If published, this will include your full peer review and any attached files.

Reviewer #1: No

Reviewer #2: No

---

## [Author Response · Author response to Decision Letter 0]

9 May 2024

Dr. Muhammad Shahzad Aslam

Academic Editor 

PLOS ONE

9th May 2024

Dear Dr Shahzad Aslam, 

Re: Cognitive Behaviour Therapy for Depersonalisation Derealisation Disorder (CBT-f-DDD): Study Protocol for a Randomised Controlled Feasibility Trial

Thank you for your kind email of the 25th of March 2024.

We are delighted to respond to the reviewer comments raised during review and we thank you for having considered our submission. We are grateful to the reviewer for raising these interesting points which we are hopeful that we have addressed satisfactorily in this review.

We identify changes to the text we have made in the main submission here in blue for ease of reference. 

We would be delighted to reconsider any further reviewer comments if necessary. 

Kindest regards, 

Elaine Hunter

e.hunter@ucl.ac.uk

Department of Psychiatry,

University College London, UK

Reviewer Comment

1. Sample Size Justification: While the protocol outlines the rationale for the chosen sample size based on feasibility, further justification regarding its adequacy for preliminary efficacy analysis might strengthen the proposal. Exploring the expected effect size based on prior studies could offer additional insight into the chosen sample size.

Response

There is no formal sample size calculation, or priori power calculation. Instead, a minimum of 24 participants, 12 in each arm, has been suggested for pilot studies. This minimum recruitment strategy requires recruiting 29 eligible participants in total (encompassing a nominal 20% dropout rate). This study aims to recruit 20 participants in each arm who complete. We believe this number will give us a final sample large (and sufficiently diverse) enough to assess the practicalities of recruitment, retention, randomisation, and intervention delivery, as well as to obtain the necessary data for the statistical inferences required. 

A previous naturalistic self-controlled cross-over analysis found the effect size between the treatment period and the waiting period to be 0.52 (improvement in favour of treatment). This translated into a decrease of 36 points on the CDS Scale. Using this change score, a correlation of 0.9 which was observed for intra-participant correlation, and the standard deviation of the CDS at each phase of the crossover study, we anticipated that the required sample size would be 29 participants per arm (using the ANCOVA method). Our planned recruitment of 20 per arm provides sufficient power to provide information on the magnitude, direction and precision of the treatment effect from the feasibility study to power (with sufficient precision) a subsequent definitive RCT if appropriate. (Hunter ECM, Wong CLM, Gafoor R, Lewis G, David AS. Cognitive Behaviour Therapy (CBT) for Depersonalization Derealization Disorder (DDD): a self-controlled cross-over study of waiting list vs. active treatment. Cogn Behav Ther. 2023 Nov;52(6):672-685.)

Reviewer Comment

2. Diversity and Inclusion: Given the mention of narrow population demographics in previous studies, it would be beneficial to outline specific strategies for ensuring a diverse participant pool, addressing potential barriers to participation for underrepresented groups.

Response

Diversity and Inclusion. Diversity and Inclusion. To increase the likelihood that the study sources a representative sample, recruitment will take place across two London mental health trusts covering six boroughs. Camden and Islington NHS Foundation Trust (C&I) is an inner London trust which had over 45,600 service users in 2020 (18). The trust covers the boroughs of Camden, Islington, and Kingston. Islington is the most densely populated local authority area in England and Wales and is among the poorest 20% of neighbourhoods in England (19). Barnet, Enfield and Haringey NHS Foundation Trust (BEH) is an outer London trust which had 146,000 service users in 2022 (20). Approximately a third of individuals across all six boroughs are residents from black, Asian or minority ethnic groups (19, 21–25). We will be recruiting from primary, secondary and tertiary care services across these Trusts.

Reviewer Comment

3. Clarification on Control Group Treatment: The protocol mentions that the control group will receive Treatment as Usual (TAU). Clarifying what TAU entails within the two NHS trusts and how it differs from the intervention could provide valuable context for interpreting future results.

Response

For some participants, the TAU will involve contact with a therapist through their mental health team or medication prescribed by their GP and/or mental health team. The TAU will vary from individual to individual but may consist of waitlist, short-term psychological therapies, or prescription of antidepressants such as SSRIs. Participants will be asked to keep a record of the type and frequency of their appointments with clinical services and treatments offered. These data will be supplemented with information provided by their key worker or through clinical note screening.

Reviewer Comment

4. Long-Term Follow-Up Plans: Although the study focuses on short-term feasibility and acceptability, discussing plans or potential for long-term follow-up could enhance understanding of the sustained impact of CBT-f-DDD, addressing whether symptom improvements are maintained over time.

Response

Unfortunately, due to the trial being funded as a feasibility study there is no scope for longer-term follow-up plans. This would be an important aspect to address if a subsequent definitive trial was funded. 

Reviewer Comment

5. Dissemination Strategy: Expanding on the dissemination plans to include patient and public involvement strategies might increase the study's impact. Outlining how findings will be communicated to participants, patient groups, and the wider public could foster engagement and awareness.

Response

Dissemination of the findings will take place within Unreal, NHS partners, University College London (UCL), and the wider academic/CBT community. This will be achieved through conference presentations, and written peer-reviewed publications. Participants will have the option of requesting study results upon completion by ticking a box in the consent form. Unreal as a co-applicant on this study will assist the dissemination of materials to the lay public across various platforms. There will be a plain English summary published on the charity website and the results of the study will be included in their charity newsletter which is circulated to 1100 recipients (33). Additionally, a plain English thread will be posted on their X account to reach both wider academic audiences and clinical populations. 

Reviewer Comment

6. Adjust citation method to adhere to journal guidelines.

Response

This has been adjusted to adhere to journal guidelines.

Reviewer Comment

7. Clarify the term "talking therapy" and specify included treatments like counselling or CBT.

Response

The wording has been adjusted to psychological therapies. 

Reviewer Comment

8. Include the number of sessions in the initial CBT study on DDD.

Response

This was included in text: 

There have been two uncontrolled studies of consecutive referrals of patients with DDD to a tertiary specialist clinic 10,11. In the first of these, 21 participants, most with longstanding severe symptoms which had not responded to medication or talking therapies, were treated with a median of 11 individual CBT sessions (IQR: 7 to 18). There was a statistically significant reduction in state DDD symptom severity (Effect Size = 0.37, 95%; [C.I: 0.12 to 0.53]) with 29% of participants no longer meeting diagnostic criteria at the end of therapy. Additionally, participants experienced a reduction in the severity of general dissociation (Effect Size = 0.34; [95% C.I: 0.10 to 0.51]). A recent analysis of outcomes of 36 participants from the same clinic has replicated these initial findings 11. A mean decrease in trait DDD symptoms (Mean = -33.67; [95% C.I: -45.14 to -22.20]) was observed after a median of 16 sessions of CBT (IQR: 10 to 16), with additional statistically significant reductions in anxiety and depression symptoms during the treatment period relative to the waitlist period.

Reviewer Comment

9. Expand literature review to cover various DDD treatments comprehensively.

Response

In systematic reviews of the existing evidence of treatment for DDD, the strongest evidence is for CBT that has been adapted for the specific condition, with little benefit demonstrated from a variety of medications and other therapy modalities. Wang et al.,’s (9) recent systematic review comprehensively covered both psychiatric and therapeutic modalities for treating DDD concluding that the evidence was poor, heterogenous and cannot support clinical treatment. The main treatment modalities discussed included medication primarily the use of lamotrigine and Selective Serotonin Reuptake Inhibitors (SSRIs) (11–13); neuromodulation of the temporo-parietal junction and ventrolateral prefrontal cortex through repetitive transcranial magnetic stimulation (14); and CBT methods as described in this study (15–17). 

Reviewer Comment

10. Elaborate on the process of examining diagnostic criteria, including details on diagnostic interviews and personnel involved.

Response

The study will aim to recruit 40 participants, 20 participants in each arm. All participants will be aged between 16-75, living in London, and have symptoms of Depersonalisation and/or Derealisation currently meeting DSM-V (1) criteria for DDD. This will be assessed by the research assistant in their initial screening interview through asking the participant to the following two questions, which make up the diagnostic criteria as set out in the DSM-V (1): 

“Over the last two weeks, how often have you had: 

a) Experiences of unreality, detachment, or being an outside observer with respect to one's thoughts, feelings, sensations, body, or actions (e.g., perceptual alterations, distorted sense of time, unreal or absent self, emotional and/or physical numbing). [Depersonalisation] 

0 =not at all; 1=several days; 2=more than half the days; 3=nearly every day”

b) Experiences of unreality or detachment with respect to surroundings (e.g., individuals or objects are experienced as unreal, dreamlike, foggy, lifeless, or visually distorted). [Derealisation]

0=not at all; 1=several days; 2=more than half the days; 3=nearly every day.”

The total score will be calculated as follows: 

1. Total score = (a + b). 

a. Scores <3 = mild-moderate severity DDD. 

b. Scores >=3 = moderate-severe severity DDD.

Other DSM-V criteria will be assessed including intact reality testing, that the symptoms result in significant distress and/or functional impairment, and that the DDD symptoms cannot be better explained by another condition. Therefore, participants will be excluded if they have any of the following co-morbid diagnoses: psychosis spectrum disorder, substance dependence, intellectual disability, Post Traumatic Stress Disorder (PTSD), or cognitive impairment due to head injury or organic disorder. Comorbidity with anxiety and/or depressive disorders will not be an exclusion criterion.

Reviewer Comment

11. Provide more information about "treatment as usual."

Response

For some participants, the TAU will involve contact with a therapist through their mental health team or medication prescribed by their GP and/or mental health team. The TAU will vary from individual to individual but may consist of waitlist, short-term psychological therapies, or prescription of antidepressants such as SSRIs. Participants will be asked to keep a record of the type and frequency of their appointments with clinical services and treatments offered. These data will be supplemented with information provided by their key worker or through clinical note screening.

Reviewer Comment

12. Specify the qualitative method used for data analysis.

Response

Qualitative data from interviews with therapists and participants will be transcribed and analysed using thematic analysis within NVivo software (supervised by co-applicant Dr Morant). This will be guided by Braun & Clarke’s six stage process (42), and conducted collaboratively to include the perspectives of clinicians and those with lived experience within the research team. Interviews will be recorded using an encrypted device for the audio recording, and data will be stored securely within the NHS. Transcription of this audio recording will be done by a professional external agency which has an over-arching contract with UCL that covers all GDPR issues (as outlined in the collaboration agreement between UCL and C&I). These audio files will be destroyed at the end of the project. 

Reviewer Comment

13. Discuss measures taken to compare in-person and online therapy methods, considering potential differences and limitations.

Response

The location of these sessions will depend on participant and/or therapist preference and can be conducted face-to-face or via videoconferencing. Meta-analyses comparing the efficacy of psychological therapies conducted face-to-face versus through videoconferencing has shown similar symptomatic improvement and client satisfaction scores across conditions (21,22). Additionally, a recent RCT examining online versus in-person CBT for anxiety conditions did not find a significant difference in outcome measures between treatment conditions(23). It is not expected that this study will find any differences between the two conditions but will collect the modality used for monitoring purposes.

Reviewer Comment

14. Explain the duration of treatment sessions.

Response

Those in the intervention arm will receive a minimum of 12 (maximum of 24) 1-hour-long CBT-f-DDD sessions.

---

## [Decision Letter · Decision Letter 1]

5 Jun 2024

PONE-D-24-01397R1Cognitive Behaviour Therapy for Depersonalisation Derealisation Disorder(CBT-f-DDD): Study Protocol for a Randomised Controlled Feasibility TrialPLOS ONE

Dear Dr. Hunter,

Thank you for submitting your manuscript to PLOS ONE. After careful consideration, we feel that it has merit but does not fully meet PLOS ONE’s publication criteria as it currently stands. Therefore, we invite you to submit a revised version of the manuscript that addresses the points raised during the review process.

The reviewer agrees with the author that calculating the sample size for a preliminary efficacy analysis is unnecessary because the study is exploratory. The sample size is too small to test efficacy, so statistical analysis should avoid hypothesis testing.

The reviewer suggests removing the aim of characterizing the probability that the results were obtained by chance from the text. They explain that while a significant p-value might indicate results are unlikely due to chance, it does not accurately measure this probability as it is influenced by many factors.

They recommend interpreting p-values with caution within the context of the study design and data. For a pilot study focused on testing feasibility, they advise against conducting hypothesis testing and reporting p-values.

We look forward to receiving your revised manuscript.

Kind regards,

Muhammad Shahzad Aslam, Ph.D.,M.Phil., Pharm-D

Academic Editor

PLOS ONE

Journal Requirements:

Reviewers' comments:

Reviewer's Responses to Questions

**Comments to the Author**

1. Does the manuscript provide a valid rationale for the proposed study, with clearly identified and justified research questions?

Reviewer #1: Yes

Reviewer #3: Yes

2. Is the protocol technically sound and planned in a manner that will lead to a meaningful outcome and allow testing the stated hypotheses?

Reviewer #1: Yes

Reviewer #3: Yes

3. Is the methodology feasible and described in sufficient detail to allow the work to be replicable?

Reviewer #1: Yes

Reviewer #3: Yes

4. Have the authors described where all data underlying the findings will be made available when the study is complete?

Reviewer #1: Yes

Reviewer #3: No

5. Is the manuscript presented in an intelligible fashion and written in standard English?

Reviewer #1: Yes

Reviewer #3: Yes

6. Review Comments to the Author

You may also provide optional suggestions and comments to authors that they might find helpful in planning their study.

Reviewer #1: Dear Sir or Madam,

I read the revised version of the manuscript. Your dedicated effort is admirable.

I wish you success

Reviewer #3: I agree with the author that calculating the sample size for a preliminary efficacy analysis is unnecessary due to the exploratory nature of the study. The sample size is insufficient to test efficacy, and the statistical analysis should not include hypothesis testing.

The aim of characterizing the probability that the results were obtained by chance was mentioned in several places. I recommend omitting this aim completely from the text. A significant p-value may suggest that the results are unlikely to be due to chance. However, it does not characterize the probability that the results are due to chance as the p value is affected by many factors.

P value should be interpreted with caution and in the context of the overall study design and data. For a pilot study and the primary aim is testing feasibility, I would not recommend to conduct hypothesis testing and report p values.

7. PLOS authors have the option to publish the peer review history of their article (what does this mean?). If published, this will include your full peer review and any attached files.

Reviewer #1: No

Reviewer #3: No

---

## [Author Response · Author response to Decision Letter 1]

19 Jun 2024

Dr. Muhammad Shahzad Aslam

Academic Editor 

PLOS ONE

19th June 2024

Dear Dr Shahzad Aslam, 

Re: Cognitive Behaviour Therapy for Depersonalisation Derealisation Disorder (CBT-f-DDD): Study Protocol for a Randomised Controlled Feasibility Trial

Thank you for your kind email of the 6th of June 2024.

We are delighted to respond to the reviewer comments raised during review and we thank you for having considered our submission. We are grateful to the reviewer for raising these interesting points which we are hopeful that we have addressed satisfactorily in this review.

To facilitate review of our submission, we have taken the liberty of listing the comments raised in the last review and appending the changes we have made in the text (where appropriate) of the proposed manuscript.

1. The reviewer agrees with the author that calculating the sample size for a preliminary efficacy analysis is unnecessary because the study is exploratory. The sample size is too small to test efficacy, so statistical analysis should avoid hypothesis testing.

We are delighted that the reviewer concurs with our decision not to conduct a power calculation for the feasibility study.

2. The reviewer suggests removing the aim of characterizing the probability that the results were obtained by chance from the text. They explain that while a significant p-value might indicate results are unlikely due to chance, it does not accurately measure this probability as it is influenced by many factors.

We have removed the section of the text which refers to characterising the probability that the results have been obtained by chance. The text now reads:

“To carry out preliminary inferential analyses of the efficacy of the therapy to check if a signal has been detected in the treatment arm versus the control arm (so as to inform a subsequent larger RCT if appropriate). These analyses may include estimating the magnitude and direction of the treatment effect as well as obtaining an estimate of the uncertainty in the estimate. “

3. They recommend interpreting p-values with caution within the context of the study design and data. For a pilot study focused on testing feasibility, they advise against conducting hypothesis testing and reporting p-values.

We will not report p values but instead characterize the uncertainty in the estimate of the treatment effect by declaring the 95% confidence interval and standard error of the mean.

We have identified changes to the text we have made in blue for ease of reference. 

We would be delighted to reconsider any further reviewer comments if necessary. 

Kindest regards, 

Elaine Hunter

e.hunter@ucl.ac.uk

Department of Psychiatry,

University College London, UK

---

## [Decision Letter · Decision Letter 2]

2 Jul 2024

Cognitive Behaviour Therapy for Depersonalisation Derealisation Disorder(CBT-f-DDD): Study Protocol for a Randomised Controlled Feasibility Trial

PONE-D-24-01397R2

Dear Dr. Hunter,

We’re pleased to inform you that your manuscript has been judged scientifically suitable for publication and will be formally accepted for publication once it meets all outstanding technical requirements.

Kind regards,

Muhammad Shahzad Aslam, Ph.D.,M.Phil., Pharm-D

Academic Editor

PLOS ONE

Additional Editor Comments (optional):

Reviewers' comments:

Reviewer's Responses to Questions

**Comments to the Author**

1. Does the manuscript provide a valid rationale for the proposed study, with clearly identified and justified research questions?

Reviewer #3: Yes

2. Is the protocol technically sound and planned in a manner that will lead to a meaningful outcome and allow testing the stated hypotheses?

Reviewer #3: Yes

3. Is the methodology feasible and described in sufficient detail to allow the work to be replicable?

Reviewer #3: Yes

4. Have the authors described where all data underlying the findings will be made available when the study is complete?

Reviewer #3: Yes

5. Is the manuscript presented in an intelligible fashion and written in standard English?

Reviewer #3: Yes

6. Review Comments to the Author

You may also provide optional suggestions and comments to authors that they might find helpful in planning their study.

Reviewer #3: It is appropriate to report efficacy magnitude and 95% confidence interval.

All my comments are answered.

7. PLOS authors have the option to publish the peer review history of their article (what does this mean?). If published, this will include your full peer review and any attached files.

Reviewer #3: No

---

## [Editor Report · Acceptance letter]

1 Aug 2024

PONE-D-24-01397R2 

PLOS ONE

Dear Dr. Hunter, 

I'm pleased to inform you that your manuscript has been deemed suitable for publication in PLOS ONE. Congratulations! Your manuscript is now being handed over to our production team.

Kind regards, 

on behalf of

Dr. Muhammad Shahzad Aslam 

Academic Editor

PLOS ONE